# Antimicrobial Susceptibility Testing Using the MYCO Test System and MIC Distribution of 8 Drugs against Clinical Isolates of Nontuberculous Mycobacteria from Shanghai

Ruoyan Ying,[a,b] Jinghui Yang,[c] Xiaocui Wu,[c] Fangyou Yu,[c] Wei Sha[a,b]

[a]Shanghai Key Laboratory of Tuberculosis, Shanghai Pulmonary Hospital, Tongji University School of Medicine, Shanghai, People's Republic of China
[b]Tuberculosis Center for Diagnosis and Treatment, Shanghai Pulmonary Hospital, Tongji University School of Medicine, Shanghai, People's Republic of China
[c]Clinical Laboratory, Shanghai Pulmonary Hospital, Tongji University School of Medicine, Shanghai, People's Republic of China

Ruoyan Ying and Jinghui Yang contributed equally to this work. The authors' order was determined by work load.

**ABSTRACT** Given the increased incidence and prevalence of nontuberculous mycobacterial (NTM) diseases and the natural resistance of NTM to multiple antibiotics, *in vitro* susceptibility testing of different NTM species against drugs from the MYCO test system and new applied drugs is required. A total of 241 NTM clinical isolates were analyzed, including 181 slowly growing mycobacteria (SGM) and 60 rapidly growing mycobacteria (RGM). The Sensititre SLOMYCO and RAPMYCO panels were used for testing susceptibility to commonly used anti-NTM antibiotics. Furthermore, MIC distributions were determined against 8 potential anti-NTM drugs, including vancomycin (VAN), bedaquiline (BDQ), delamanid (DLM), faropenem (FAR), meropenem (MEM), clofazimine (CLO), cefoperazone-avibactam (CFP-AVI), and cefoxitin (FOX), and epidemiological cutoff values (ECOFFs) were analyzed using ECOFFinder. The results showed that most of the SGM strains were susceptible to amikacin (AMK), clarithromycin (CLA), and rifabutin (RFB) from the SLOMYCO panels and BDQ and CLO from the 8 applied drugs, while RGM strains were susceptible to tigecycline (TGC) from the RAPMYCO panels and also BDQ and CLO. The ECOFFs of CLO were 0.25, 0.25, 0.5, and 1 $\mu$g/mL for the mycobacteria *M. kansasii*, *M. avium*, *M. intracellulare*, and *M. abscessus*, respectively, and the ECOFF of BDQ was 0.5 $\mu$g/mL for the same four prevalent NTM species. Due to the weak activity of the other 6 drugs, no ECOFF was determined. This study on the susceptibility of NTM includes 8 potential anti-NTM drugs and a large sample size of Shanghai clinical isolates and demonstrates that BDQ and CLO had efficient activities against different NTM species *in vitro*, which can be applied to the treatment of NTM diseases.

**IMPORTANCE** We designed customized panel that contains 8 repurposed drugs, including vancomycin (VAN), bedaquiline (BDQ), delamanid (DLM), faropenem (FAR), meropenem (MEM), clofazimine (CLO), cefoperazone-avibactam (CFP-AVI), and cefoxitin (FOX) from the MYCO test system. To better understand the efficacy of these 8 drugs against different NTM species, we determined the MICs of 241 NTM isolates collected in Shanghai, China. We attempted to define the tentative epidemiological cutoff values (ECOFFs) for the most prevalent NTM species, which is an important factor in setting up the breakpoint for a drug susceptibility testing. We used the MYCO test system as an automatic quantitative drug sensitivity test of NTM and extended the method to BDQ and CLO in this study. The MYCO test system complements commercial microdilution systems that currently lack BDQ and CLO detection.

**KEYWORDS** nontuberculous mycobacteria, drug susceptibility, bedaquiline, clofazimine, slowly growing mycobacteria, rapidly growing mycobacteria, epidemiological cutoff values

Address correspondence to Wei Sha, shfksw@126.com, or Fangyou Yu, wzjxyfy@163.com.

The authors declare no conflict of interest.

Infections caused by nontuberculous mycobacteria (NTM) are increasing worldwide (1–3), which is becoming a major new global health issue. Most NTM are intrinsically resistant or only partially sensitive to first-line antituberculosis (anti-TB) drugs. Therefore, treatment of NTM disease usually takes a very long period of time, with a regimen consisting of a macrolide as the core drug and 2 to 4 other antibiotics until 12 months after sputum conversion (4–6).

Despite the increasing number of patients infected with NTM, the cure rate of NTM pulmonary disease (NTM-PD) remains unsatisfactory (7–9). Given the situation that the process of research and development of potent novel antibiotics specific to NTM is sluggish, evaluation of the effects of existing antibiotics and new antituberculosis drugs (here, we call them "repurposed drugs") on NTM is a feasible way to develop new therapeutic strategies for NTM-PD.

NTM are Gram-positive, aerobic bacteria belonging same genus as *Mycobacterium tuberculosis*. $\beta$-Lactam antibiotics, like carbapenem, are introduced to treat some NTM infections, with or without $\beta$-lactamase (5, 6, 10, 11). In recent years, new generations of $\beta$-lactam antibiotics and oral preparations have attracted attention since preclinical studies and case reports presented promising efficacy in treating infections due to NTM (12–14). Meanwhile, new anti-TB drugs, such as bedaquiline (BDQ), delamanid (DLM), and clofazimine (CLO), have been shown potent bacteriostatic activity against NTM both *in vivo* and *in vitro* (15–17). Guidelines recommend a "culture-species identification-drug susceptibility test (DST)" procedure before treatment is initiated: it is important to conduct *in vitro* a DST to predict the therapeutic effects of these repurposed drugs. In previous studies, relatively small numbers of clinical isolates were tested and few studies observed drug sensitivity to both $\beta$-lactams and novel antituberculosis drugs simultaneously.

The DST recommended by CLSI is broth microdilution using antimicrobial concentrations derived from serial 2-fold dilutions indexed to the base 2. In this study, we used commercial products of MYCO test system, which are microdilution assays containing lyophilized commonly used anti-NTM antibiotics, for the MIC determination of slowly growing mycobacteria (SGM) (Sensititre SLOMYCO) and rapidly growing mycobacteria (RGM) (Sensititre RAPMYCO). At the same time, we designed a customized panel that included 8 repurposed drugs: vancomycin (VAN), bedaquiline (BDQ), delamanid (DLM), faropenem (FAR), meropenem (MEM), clofazimine (CLO), cefoperazone-avibactam (CFP-AVI), and cefoxitin (FOX) to evaluate the *in vitro* antimicrobial activity of these drugs against different NTM species.

## RESULTS

**MIC distribution in MYCO test system using reference strains.** Details of the 241 NTM clinical isolates analyzed in this study are shown in Fig. 1 and discussed further below.

For the reference strains of *Staphylococcus aureus* (ATCC 29213), *Escherichia coli* (ATCC 35218), and *Mycobacterium avium* (ATCC 19420), the results were concordant for all drugs recommended by CLSI (18). H37RV (ATCC 27294) was used for quality control of DLM.

**MIC distribution of SGM species in SLOMYCO panel.** Amikacin (AMK), with a $MIC_{50}$ of 8 $\mu$g/mL and $MIC_{90}$ of 16 $\mu$g/mL, clarithromycin (CLA), with a $MIC_{50}$ of 1 $\mu$g/mL and $MIC_{90}$ of 2 $\mu$g/m, and rifabutin (RFB), with a $MIC_{50}$ of 0.5 $\mu$g/mL and $MIC_{90}$ of 1 $\mu$g/mL, had strong sensitivity to SGM species as shown in Table 1. By further stratified analysis, the rates of MIC $\leq$ breakpoint for susceptibility for *Mycobacterium intracellulare* to the 6 antimicrobial agents AMK, CLA (macrolide antibiotics), ethambutol (EMB), linezolid (LZD), moxifloxacin (MXF), and RFB were 90.2% (110/122), 95.9% (117/122), 96.7% (118/122), 4.9% (6/122), 6.6% (8/122), and 96.7% (118/122), respectively, based on the breakpoints for susceptibility suggested by the Clinical and Laboratory Standards Institute (CLSI) (19) or established for many of the drug-organism combinations tested in this study. No breakpoint was determined for ciprofloxacin (CIP), doxycycline (DOX),

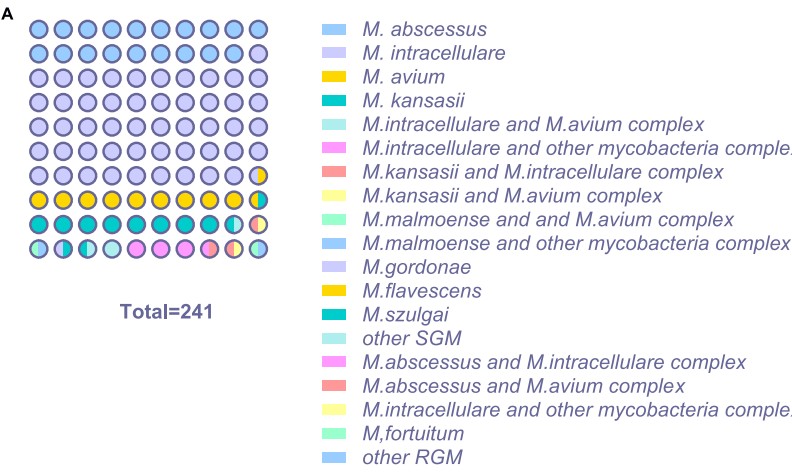

**FIG 1** Strain category of 241 clinical isolates in this study.

ethyl ethionamide (ETH), isoniazid (INH), rifampin (RIF), streptomycin (STR), and trimethoprim-sulfamethoxazole (SXT), due to the weak activity against *M. intracellulare*.

The rates of MIC ≤ breakpoint for susceptibility for *M. avium* to the same 13 antimicrobial agents were 100% (24/24), 87.5% (21/24), 100% (24/24), 91.7% (22/24), 79.2% (19/24), 37.5% (9/24), 75% (18/24), 91.7% (22/24), and 75% (18/24), respectively, for AMK, CIP, CLA, ETH, LZD, MXF, RIF, RFB, and SXT. No breakpoint was determined for DOX, EMB, INH, and STR, due to the weak activity against *M. avium*. Similar to findings for *M. intracellulare*, *M. avium* showed strong sensitivity to only AMK, CLA, and RFB among the 13 antibacterial agents, but it also had certain sensitivity to ETH, LZD, and RIF.

Meanwhile, the rates of MIC ≤ breakpoint for susceptibility for *M. kansasii* for the same 13 antimicrobial agents were 95.2% (20/21), 33.3% (7/21), 95.2% (20/21), 9.5% (2/21), 42.9% (9/21), 95.2% (20/21), 100% (21/21), 100% (21/21), 100% (21/21), 100% (21/21), 100% (21/21), 81.0% (17/21), and 81.0% (17/21), respectively, for AMK, CIP, CLA, DOX, EMB, ETH, INH, LZD, MXF, RIF, RFB, STR, and SXT (breakpoint of ≤2/38 μg/mL) as shown in Fig. 2.

**MIC distribution of SGM species in 8 repurposed antimicrobial agents.** Among the 8 repurposed antibacterial drugs, the $MIC_{50}$ and $MIC_{90}$ of BDQ and CLO were ≤0.12 and 0.25 μg/mL as well as ≤0.06 and 0.125 μg/mL, respectively, which were very close to their minimum MIC range, showing excellent sensitivity. The $MIC_{90}$ values of the other 6 drugs exceeded their respective maximum MIC range as shown in Table 2. By further stratified analysis, for *M. kansasii*, 95.2% (20/21) of

**TABLE 1** Distribution of $MIC_{50}$ and $MIC_{90}$ values for 181 isolates of SGM in the SLOMYCO panel

| Antimicrobial agent | MIC (μg/mL) | | |
|---|---|---|---|
| | $MIC_{50}$ | $MIC_{90}$ | Range |
| AMK | 8 | 16 | 1–64 |
| CIP | 16 | >16 | 0.12–16 |
| CLA | 1 | 2 | 0.06–64 |
| DOX | >16 | >16 | 0.12–16 |
| EMB | 4 | 16 | 0.5–16 |
| ETH | 20 | >20 | 0.3–20 |
| INH | >8 | >8 | 0.25–8 |
| LZD | 32 | 32 | 1–64 |
| MXF | 4 | 4 | 0.12–8 |
| RIF | 8 | >8 | 0.12–8 |
| RFB | 0.5 | 1 | 0.25–8 |
| STR | 32 | >64 | 0.5–64 |
| SXT | 4 | >8 | 0.12–8 |

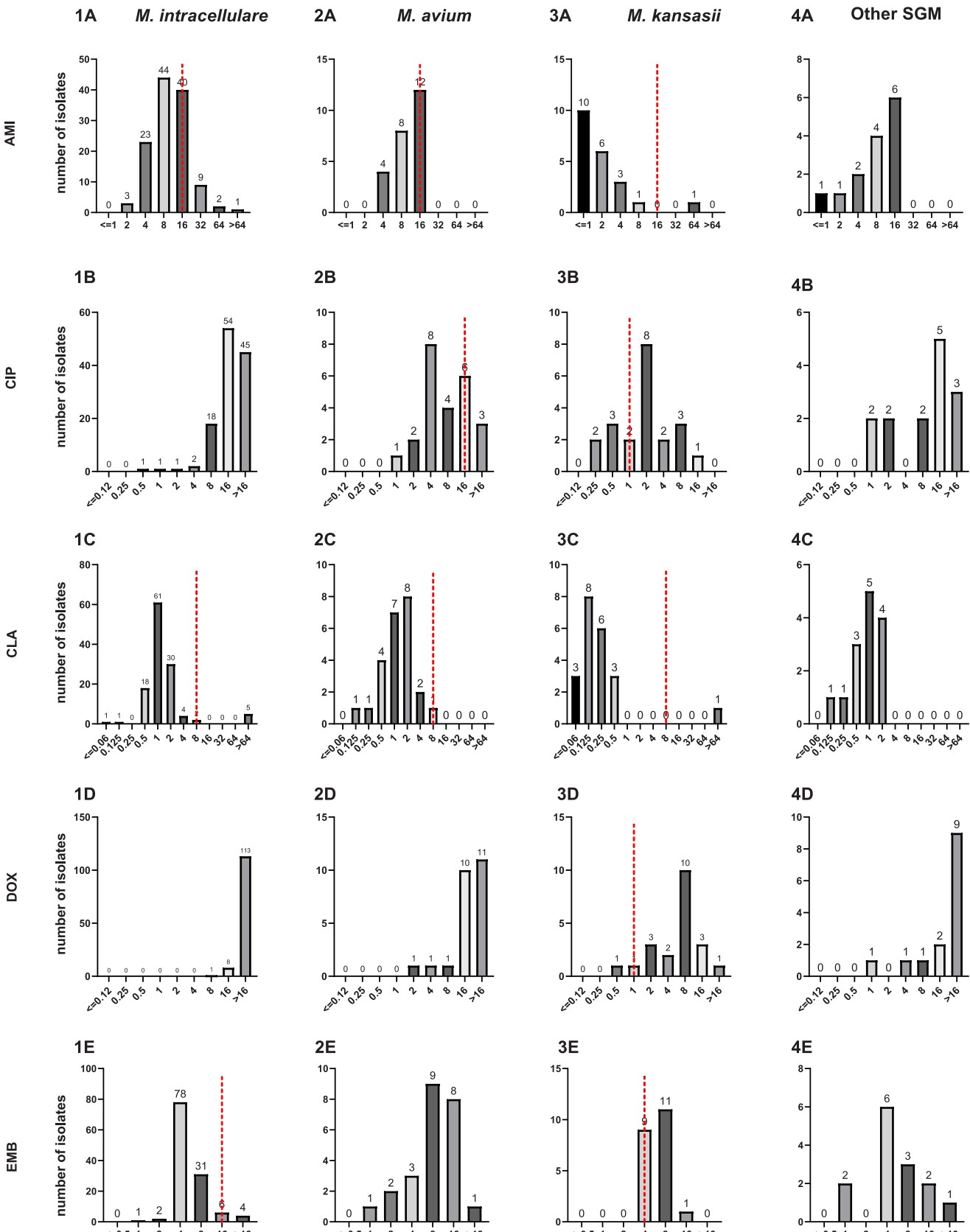

**FIG 2** MIC distribution of the 3 most prevalent NTM species in SGM in the SLOMYCO panel. Dashed lines represent the CLSI susceptibility breakpoints. The distributions of 1A to 1M, 2A to 2M, 3A to 3M and 4A to 4M represent the MIC distribution of *M. intracellulare*, *M. avium*, *M. kansasii* and other SGM in the SLOMYCO panel for 13 drugs, respctively, with the number on the bar representing the number of clinical isolates.

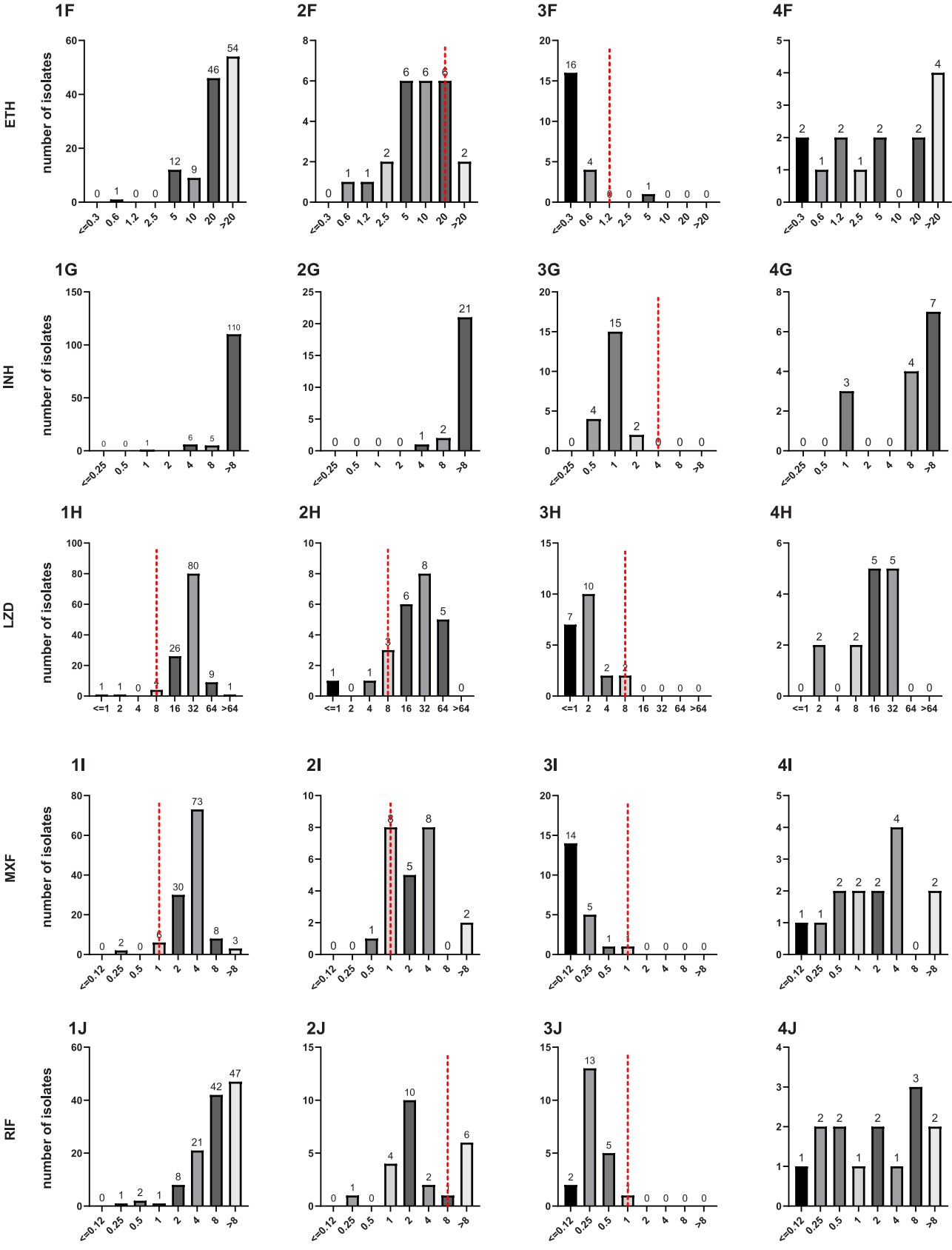

**FIG 2** (Continued)

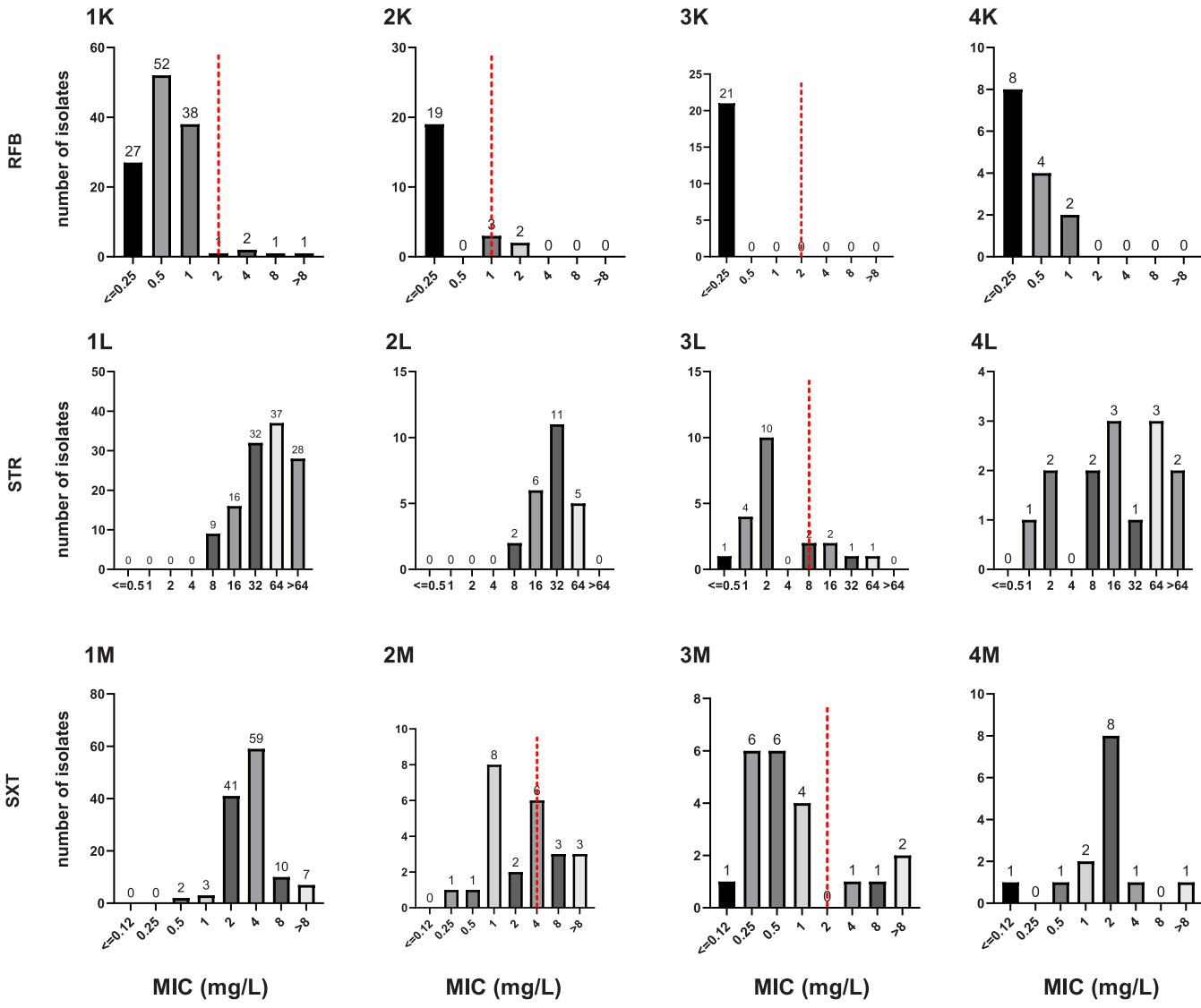

**FIG 2** (Continued)

strains had a MIC of ≤0.12 µg/mL, below the minimum MIC range for BDQ, showing excellent sensitivity. For CLO, the MIC was ≤0.06 µg/mL for all strains, which was lower than the minimum MIC range, also showing excellent sensitivity. Due to the low maximum MIC range of DLM, the number of strains below the maximum MIC range was 66.7% (14/21), showing partial sensitivity. For VAN, 52.4% (11/21) of all strains were below the maximum MIC range, for FAR the percentage was 42.9% (9/21), for MEM it was 14.3% (3/21), and for CFP-AVI it was 4.8% (1/21). For FOX, all strains had a value that was greater than the maximum MIC range, and there were no strains close to the minimum MIC range for VAN, FAR, MEM, CFP-AVI, and FOX, indicating that the bacteriostatic effects of these 5 drugs on *Mycobacterium kansasii* were limited. Similar to findings of *M. avium* and *M. intracellulare*, 99.2% (120/121) and 95.8% (23/24) of strains had MICs of ≤0.12 µg/mL, respectively, below the minimum MIC range for BDQ; 48.8% (59/121) and 45.8% (11/24) of strains had MICs of ≤0.06 µg/mL, respectively, below the minimum MIC range for CLO and showed the same sensitivity to BDQ and CLO. The MIC values of DLM, VAN, FAR, MEM, CFP-AVI, and FOX were close to their respective maximum MIC range, indicating that these 6 drugs had a poor antibacterial effect on *M. avium* and *M. intracellulare* as shown in Table 3.

**TABLE 2** Distribution of $MIC_{50}$ and $MIC_{90}$ values for 181 isolates of SGM in 8 repurposed antimicrobial agents in the self-defined panel

| Antimicrobial agent | MIC ($\mu$g/mL) | | |
| --- | --- | --- | --- |
| | $MIC_{50}$ | $MIC_{90}$ | Range |
| VAN | >16 | >16 | 0.5–16 |
| FAR | >16 | >16 | 0.5–16 |
| MEM | >16 | >16 | 0.5–16 |
| CFP-AVI | >16 | >16 | 0.5–16 |
| FOX | >128 | >128 | 8–128 |
| BDQ | ≤0.12 | 0.25 | 0.12–4 |
| CLO | ≤0.06 | 0.125 | 0.06–2 |
| DLM | >0.5 | >0.5 | 0.015–0.5 |

**MIC distribution of RGM species in RAPMYCOI panel.** AMK, with a $MIC_{50}$ of 8 $\mu$g/mL and $MIC_{90}$ of 32 $\mu$g/mL, and tigecycline (TGC), with a $MIC_{50}$ of 1 $\mu$g/mL, had strong sensitivity to RGM species, respectively, as shown in Table 4. By further stratified analysis, the rates of MIC ≤ breakpoint for susceptibility for *Mycobacterium abscessus* to the 11 antimicrobial agents AMK, CIP, CLA, DOX, LZD, MXF, SXT, FOX, imipenem (IMP), TGC, and TOB were 60.9% (28/46), 0% (0/46), 39.1% (18/46), 2.2% (1/46), 26.1% (12/46), 0% (0/46), 8.7% (4/46), 2.2% (1/46), 0% (0/46), 91.3% (42/46), and 0% (0/46), respectively.

**TABLE 3** MIC distribution of the 3 most prevalent NTM species in SGM in 8 repurposed antimicrobial agents

| Antimicrobial agent | MIC ($\mu$g/mL) for[a]: | | | |
| --- | --- | --- | --- | --- |
| | *M. intracellulare* (122) | *M. avium* (24) | *M. kansasii* (21) | Other SGM (14) |
| VAN | 4 (1)<br>>16 (121) | 16 (1)<br>>16 (23) | 2 (1)<br>8 (2)<br>16 (8)<br>>16 (10) | >16 (14) |
| FAR | 16 (1)<br>>16 (121) | >16 (24) | 2 (2)<br>8 (4)<br>16 (3)<br>>16 (12) | >16 (14) |
| MEM | >16 (122) | 1 (1)<br>>16 (23) | 2 (2)<br>8 (1)<br>>16 (18) | >16 (14) |
| CFP-AVI | >16 (122) | >16 (24) | 16 (1)<br>>16 (20) | >16 (14) |
| FOX | 64 (1)<br>128 (6)<br>>128 (115) | 128 (1)<br>>128 (23) | >128 (21) | 128 (1)<br>>128 (13) |
| BDQ | ≤0.12 (121)<br>>4 (1) | ≤0.12 (23)<br>0.25 (1) | ≤0.12 (20)<br>>4 (1) | ≤0.12 (14) |
| CLO | ≤0.06 (59)<br>0.12 (60)<br>0.25 (3) | ≤0.06 (11)<br>0.12 (11)<br>0.25 (2) | ≤0.06 (21) | ≤0.06 (9)<br>0.12 (4)<br>0.25 (1) |
| DLM | >0.5 (122) | >0.5 (24) | 0.06 (2)<br>0.12 (3)<br>0.25 (3)<br>0.5 (6)<br>>0.5 (7) | 0.25 (2)<br>>0.5 (12) |

[a]The number of isolates is in parentheses.

**TABLE 4** Distributions of $MIC_{50}$ and $MIC_{90}$ values for 60 isolates of RGM in the RAPMYCOI panel

| Antimicrobial agent | MIC ($\mu$g/mL) | | |
| --- | --- | --- | --- |
| | $MIC_{50}$ | $MIC_{90}$ | Range |
| AMK | 8 | 32 | 1–64 |
| CIP | >4 | >4 | 0.12–4 |
| CLA | 12 | >16 | 0.06–16 |
| DOX | >16 | >16 | 0.12–16 |
| LZD | 32 | >32 | 1–32 |
| MXF | >8 | >8 | 0.25–8 |
| SXT | >8 | >8 | 0.25–8 |
| AMC | >64 | >64 | 2/1–64/32 |
| FEP | >32 | >32 | 1–32 |
| FOX | 64 | >128 | 4–128 |
| CRO | >64 | >64 | 4–64 |
| IMP | >64 | >64 | 2–64 |
| MIN | >8 | >8 | 1–8 |
| TGC | 1 | >4 | 0.015–4 |
| TOB | 16 | >16 | 1–16 |

No breakpoint was determined for amoxicillin-clavulanic acid (AMC), FEP, ceftriaxone (CRO), and minocycline (MIN), due to the weak activity against *M. abscessus*, as shown in Fig. 3. This illustrated the multidrug resistance of *M. abscessus* and the limited range of bacteriostatic drug options for such infection.

**MIC distribution of RGM species in 8 repurposed antimicrobial agents.** Similar to SGM, RGM showed strong resistance to a variety of antimicrobial agents in the customized panel, except BDQ and CLO. Among the 8 repurposed antibacterial drugs, the $MIC_{50}$ and $MIC_{90}$ of BDQ and CLO were ≤0.12 and 0.25 $\mu$g/mL and 0.25 and 0.5 $\mu$g/mL, respectively, as shown in Table 5. BDQ was very close to the minimum MIC range and consistent with the SGM, showing excellent sensitivity. The $MIC_{50}$ and $MIC_{90}$ of CLO were gradually increased compared with those of SGM, indicating that the sensitivity of BDQ was better than that of CLO. The $MIC_{90}$ values of the other 6 drugs all exceeded the maximum MIC range. The rate of MIC of ≤0.12 $\mu$g/mL for BDQ was 87.0% (40/46), indicating that BDQ is a drug with strong potency *in vitro*. For CLO, the MIC for *M. abscessus* was not lower than the minimum MIC range, but the rate of MIC of ≤1 $\mu$g/mL was 95.7% (44/46), exhibiting very good inhibitory activities, as shown in Table 6.

**MICs and ECOFFs of BDQ against NTM strains.** BDQ showed strong activity against the employed clinical strains of RGM and SGM *in vitro*, as shown in Table 2 and Table 5. Both RGM species and SGM species had $MIC_{50}$ values of ≤0.12 $\mu$g/mL, and both had $MIC_{90}$ values of 0.25 $\mu$g/mL. The MIC of one strain of *M. intracellulare* and that of one strain of *M. kansasii* from the SGM were >4 $\mu$g/mL as shown in Fig. 4.

The MIC distributions of the most prevalent NTM species for BDQ are shown in Fig. 4. BDQ had similar antibacterial activities against SGM and RGM isolates of different species. Among the SGM strains employed here, BDQ showed strong activity against *M. avium, M. intracellulare, and M. kansasii*, with a putative epidemiological cutoff value (ECOFF) of 0.5 $\mu$g/mL, while the MIC values of most strains were ≤0.12 $\mu$g/mL. Among the RGM employed here, BDQ also had the same antibacterial activity against *M. abscessus*, representing a unified ECOFF. The numbers of other SGM and RGM species were small, and BDQ had the same inhibitory tendency against them, as shown in Tables S1 and S2 in the supplemental material.

**MICs and ECOFFs of CLO against NTM strains.** CLO demonstrated uniformly strong antibacterial activity against almost all of the employed SGM species, with a $MIC_{50}$ of ≤0.06 and $MIC_{90}$ of 0.125 $\mu$g/mL, as presented in Table 2. Furthermore, CLO also exhibited very potent *in vitro* activity against the recruited RGM species, with a $MIC_{50}$ of 0.25 $\mu$g/mL and $MIC_{90}$ of 0.5 $\mu$g/mL, as presented in Table 5.

The MIC distributions for CLO against the most prevalent NTM species are shown in Fig. 5. Regarding the susceptibility profile of the clinical isolates to CLO, there was a strong antibacterial activity for the majority of isolates of SGM for all included

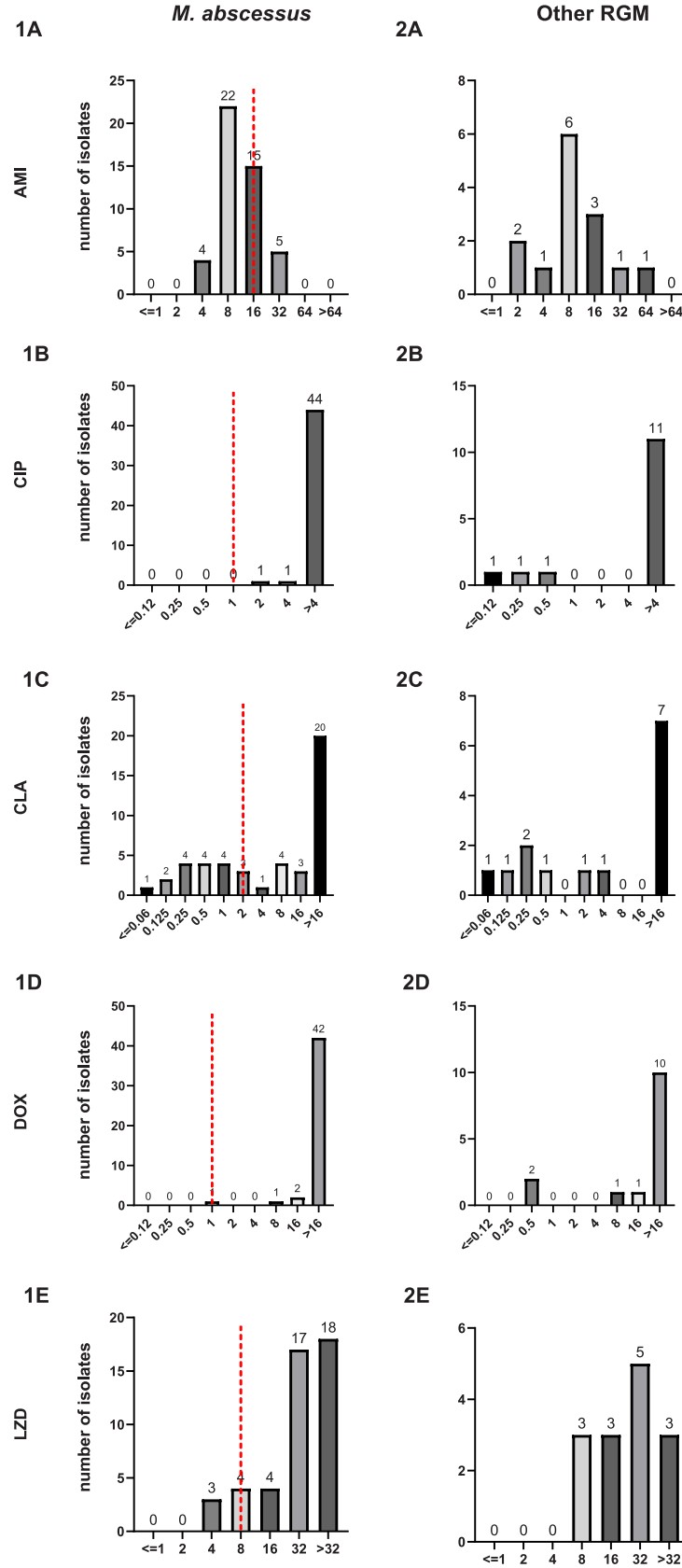

**FIG 3** MIC distribution of the most prevalent NTM species in RGM in the RAPMYCOI panel. Dashed lines represent the CLSI susceptibility breakpoints. The distributions of 1A to 1O and 2A to 2O represent the MIC distribution of *M. abscessus* and other RGM in the RAPMYCO panel for 15 drugs, respectively, with the number on the bar representing the number of clinical isolates.

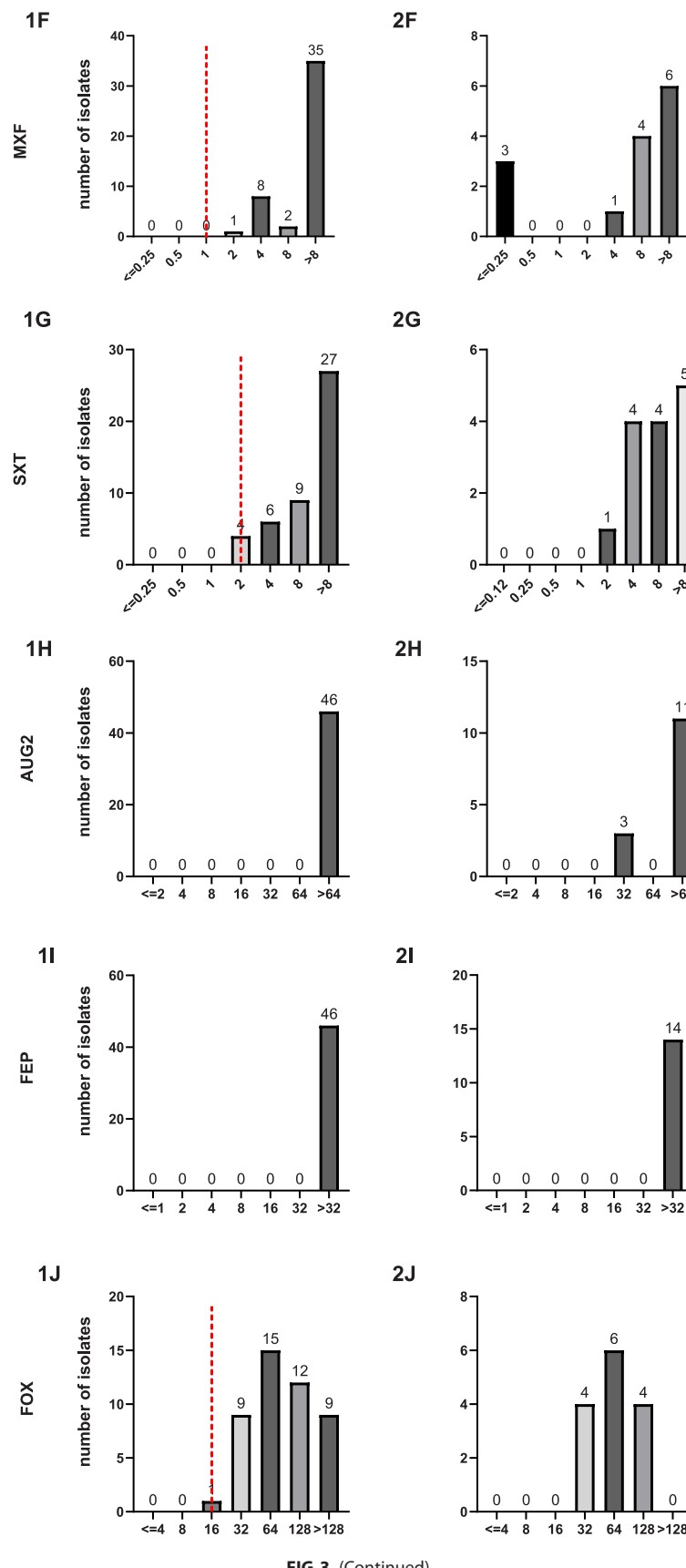

**FIG 3** (Continued)

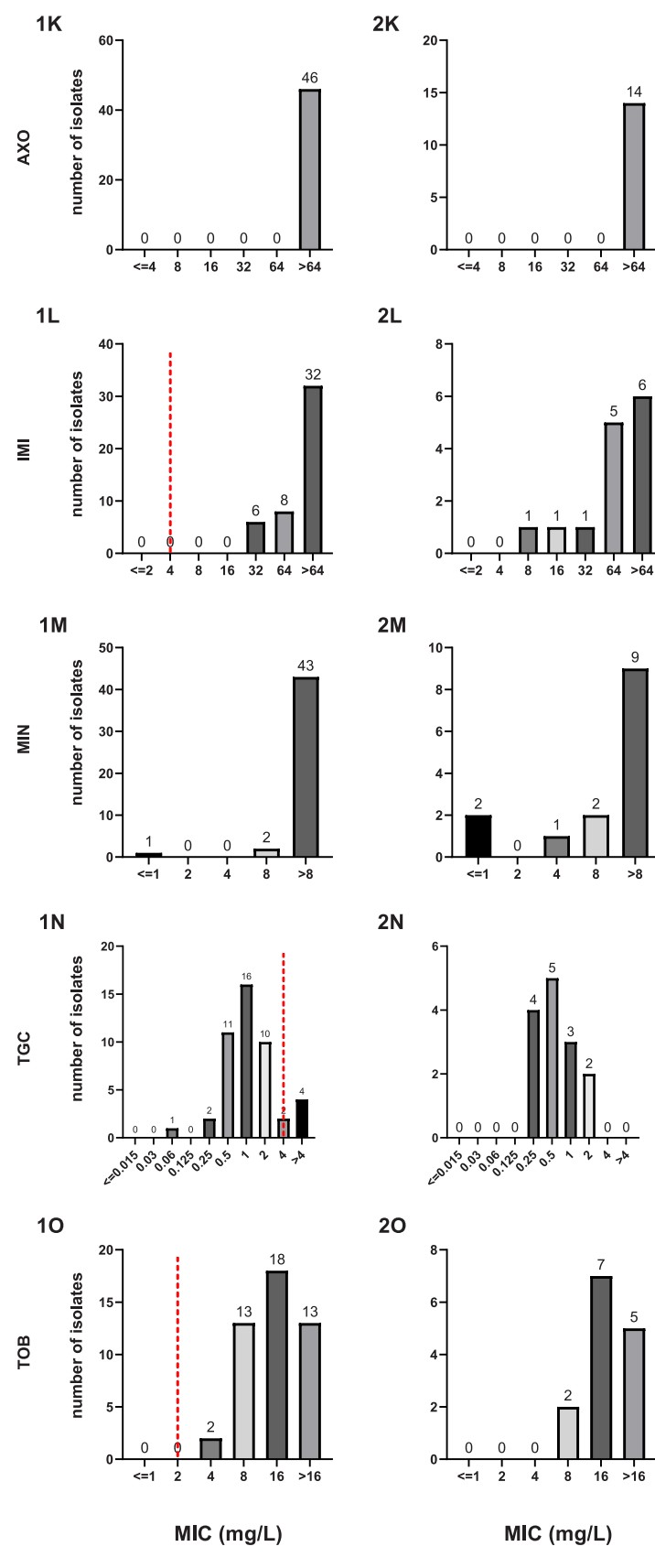

**FIG 3** (Continued)

**TABLE 5** Distribution of $MIC_{50}$ and $MIC_{90}$ values of 60 isolates of RGM in 8 repurposed antimicrobial agents in the self-defined panel

| Antimicrobial agent | MIC (µg/mL) | | |
| --- | --- | --- | --- |
| | $MIC_{50}$ | $MIC_{90}$ | Range |
| VAN | >16 | >16 | 0.5–16 |
| FAR | >16 | >16 | 0.5–16 |
| MEM | >16 | >16 | 0.5–16 |
| CFP-AVI | >16 | >16 | 0.5–16 |
| FOX | 128 | >128 | 8–128 |
| BDQ | ≤0.12 | 0.25 | 0.12–4 |
| CLO | 0.25 | 0.5 | 0.06–2 |
| DLM | >0.5 | >0.5 | 0.015–0.5 |

species. Similar activities were demonstrated against the included RGM isolates of different species, but with higher MIC values than for SGM, as shown in Fig. 5 and Table 6. Among the recruited SGM species, CLO exhibited the strongest activity against *M. kansasii* and *M. avium*, with a single tentative ECOFF of 0.25 µg/mL. Notable, the overwhelming majority of *M. kansasii* had MICs of ≤0.06 µg/mL. CLO also demonstrated strong activity against *M. intracellulare*, with a tentative ECOFF of 0.5 µg/mL. Among the employed RGM species, CLO exhibited activities against *M. abscessus*, with a tentative ECOFF of 1 µg/mL. The numbers of other SGM and RGM species were small, and CLO had the same inhibitory tendency against them, as shown in Tables S1 and S2.

**The increased MIC value of BDQ corresponds to the increased MIC value of CLO.** The MIC values of BDQ were greater than the minimum MIC range in 11 out of

**TABLE 6** MIC distribution of the most prevalent NTM species in RGM in 8 repurposed antimicrobial agents

| Antimicrobial agent | MIC (µg/mL) for[a]: | |
| --- | --- | --- |
| | *M. abscessus* (46) | Other RGM (14) |
| VAN | 16 (3)<br>>16 (43) | >16 (14) |
| FAR | >16 (46) | 4 (1)<br>>16 (13) |
| MEM | 16 (1)<br>>16 (45) | 2 (1)<br>16 (2)<br>>16 (11) |
| CFP-AVI | >16 (46) | >16 (14) |
| FOX | 64 (9)<br>128 (18)<br>>128 (19) | 32 (1)<br>64 (5)<br>128 (6)<br>>128 (2) |
| BDQ | ≤0.12 (40)<br>0.25 (3)<br>0.5 (2)<br>1 (1) | ≤0.12 (12)<br>0.25 (1)<br>>4 (1) |
| CLO | 0.12 (16)<br>0.25 (20)<br>0.5 (7)<br>1 (1)<br>>2 (2) | 0.12 (6)<br>0.25 (7)<br>0.5 (1) |
| DLM | >0.5 (46) | >0.5 (14) |

[a]The number of isolates is in parentheses.

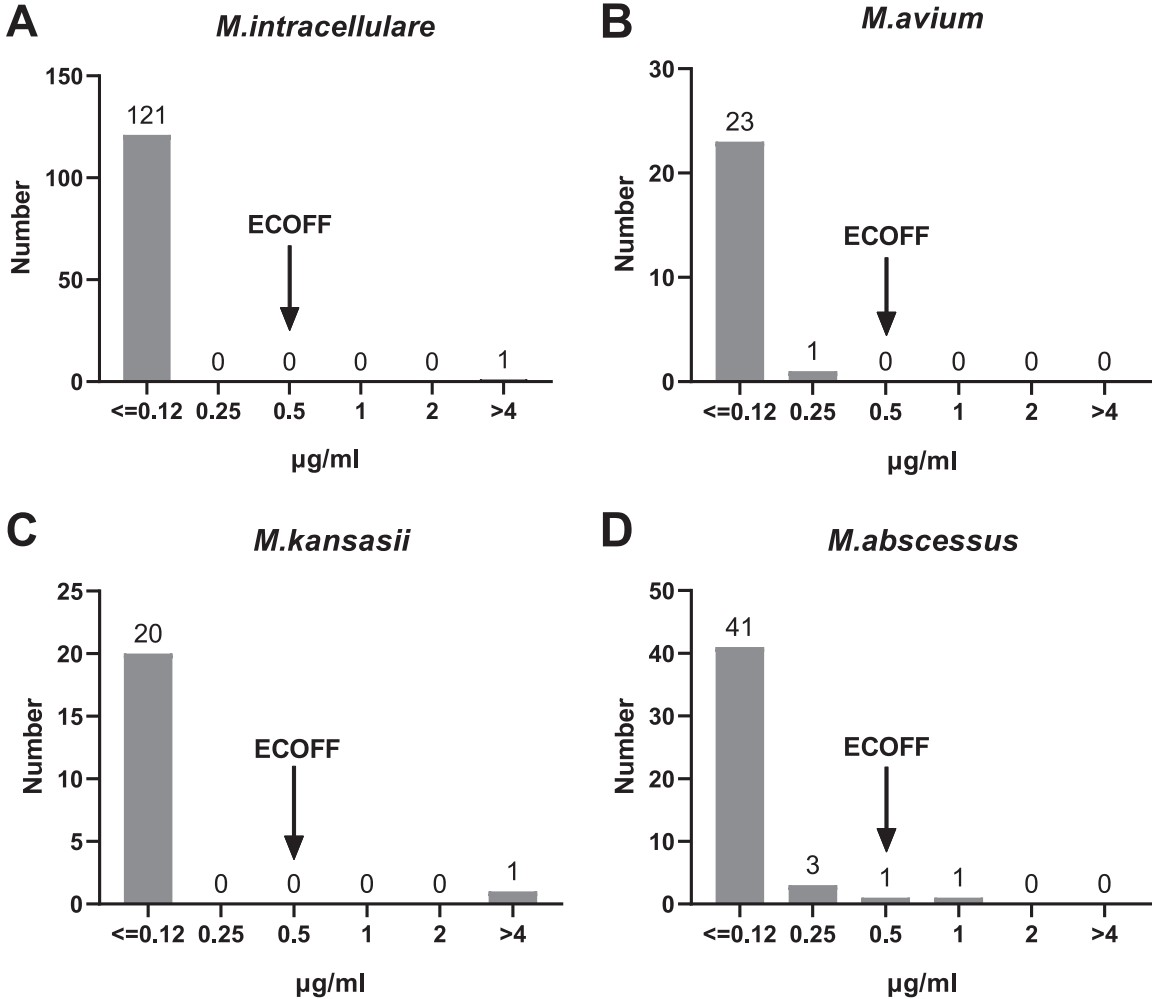

**FIG 4** MIC distributions of BDQ against the four most prevalent NTM species.

241 clinical isolates, and correspondingly, the MIC values of CLO (9/11) also increased 1- to 3-fold as shown in Table 7.

## DISCUSSION

Globally, the burden of pulmonary disease caused by NTM is increasing (20), and NTM are resistant to most commonly used antimycobacterial drugs. In this context, the off-label use of existing antibacterial drugs is being discussed as repurposed drugs are investigated. In this study, further drug sensitivity tests of VAN, BDQ, DLM, FAR, MEM, CLO, CFP-AVI, and FOX were performed on the basic efficacy of the most commonly used antibacterial drugs against SGM and RGM. In particular, BDQ, DLM, and CLO already have some clinical value in the treatment of NTM (21–25), but sensitivity and resistance criteria have not been established. We first evaluated the activity of these antimicrobials against 241 clinical NTM isolates collected from mainland China to understand their potential use against specific NTM species.

In the SLOMYCO panels, SGM was sensitive to AMK, CLA, and RFB. *M. kansasii* was more sensitive to commonly used antibacterial agents than *M. avium* and *M. intracellulare*. In addition, CIP, ETH, LZD, MXF, RIF, and SXT were more effective at inhibiting *M. intracellulare* than *M. avium* in vitro. EMB had a better bacteriostatic effect on *M. intracellulare* than *M. avium* (26). Most SGM species showed strong resistance to a variety of antibacterial agents, with the exceptions of BDQ and CLO, in the customized panels. Among the three most prevalent SGM, compared with the levels for *M. kansasii*, the MICs of more

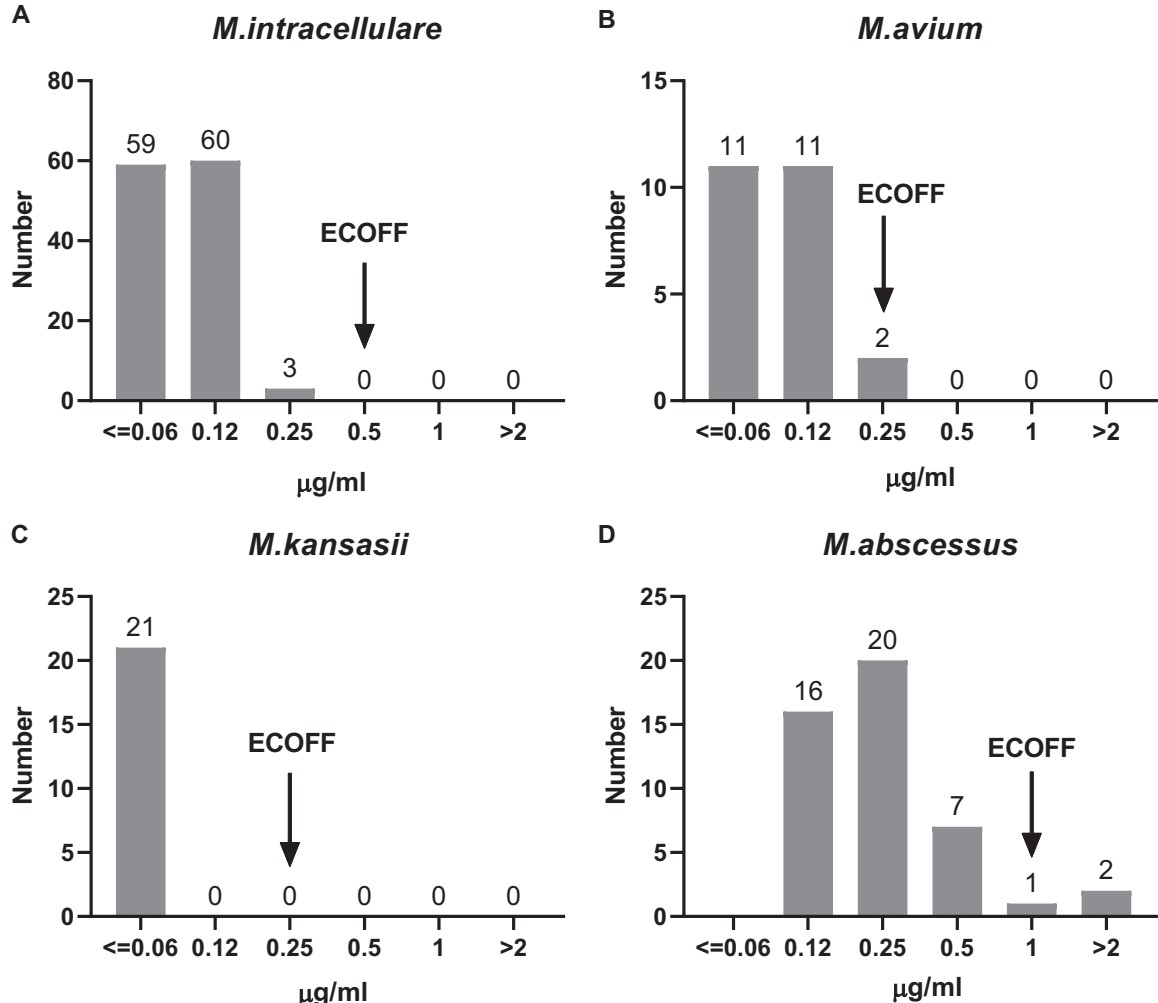

**FIG 5** MIC distributions of CLO against the four most prevalent NTM species.

than half of the strains increased 1- to 2-fold for *M. avium* and *M. intracellulare*, indicating that the latter two species are more likely to develop resistance to CLO.

Among the RAPMYCO panels, RGM was sensitive to AMK and TGC only. These results showed that RGM had strong resistance to common bacteriostatic drugs. Among the customized panels, BDQ and CLO also show excellent sensitivity in most RGM species. In addition, compared with the three most prevalent SGM, *M. kansasii*, *M. avium*, and *M. intracellulare*, the MIC of *M. abscessus* was generally increased. Treatment of RGM infection is very difficult because the bacterium is resistant to a wider variety of drugs than SGM (27).

We used the MYCO test system as an automatic quantitative drug sensitivity test of slow-growing and fast-growing NTM (28) and extended the method to BDQ and CLO in this study. It complements commercial microdilution systems that currently lack BDQ and CLO detection. The limitations of this study are high cost and the requirement for specialized equipment, but the system facilitates execution and provides more accuracy.

The breakpoints adopted in various studies were different (16, 29). Yu et al. adopted tentative ECOFFs of BDQ defined in assays for *M. kansasii*, *M. avium*, and *M. intracellulare* of ≤1 μg/mL in their study but 4 μg/mL for *M. abscessus* (16). The SLOMYCO and RAPMYCO antimicrobial susceptibility testing (AST) panel used in this study is commercially available and is more standardized and accurate than the drug-sensitive plates used in other studies. By combining the recently published data and ECOFF (30), it can be tentatively proposed that 0.5 μg/mL is the uniform breakpoint for the NTM BDQ susceptibility assay, including both RGM and SGM species. According to this

**TABLE 7** Increased MIC value of BDQ corresponding to CLO clinical isolates

| Species | MIC (µg/mL) | |
|---|---|---|
| | BDQ (range 0.12–4 µg/mL) | CLO (range, 0.06–2 µg/mL) |
| *M. kansasii* | >4 | ≤0.06 |
| *M. intracellular* and other mycobacteria complex | >4 | 0.5 |
| *M. abscessus* | 0.25 | 0.12 |
| *M. abscessus* and *M. intracellulare* complex | 0.25 | 0.25 |
| *M. avium* | 0.25 | 0.12 |
| *M. abscessus* | 0.25 | 0.25 |
| | 0.5 | 0.12 |
| | 0.5 | 0.25 |
| *M. intracellulare* | >4 | ≤0.06 |
| *M. abscessus* | 1 | 0.25 |
| | 0.25 | 0.12 |

breakpoint, the BDQ resistance rates of the four most prevalent species were 0.8% (1/122), 0% (0/24), 4.8% (1/21), and 2.2% (1/46) for *M. intracellular*, *M. avium*, *M. kansasii*, and *M. abscessus*, respectively.

By combining the ECOFF and recently published data (31), a uniform breakpoint of 1 µg/mL could be tentatively proposed for NTM CLO susceptibility testing, including both RGM and SGM species. According to this breakpoint, the CLO resistance rates of the four most prevalent species were 0% (0/122), 0% (0/24), 0% (0/21), and 4.3% (2/46) for *M. intracellular*, *M. avium*, *M. kansasii*, and *M. abscessus*, respectively. Further validation of this breakpoint is needed to support its use.

There are other important findings from this study. For example, among the 241 patients, we detected an increase in MIC values for BDQ in 11 strains, followed by an increase in MIC values of CLO (9/11), as shown in Table 7. Meanwhile, the same phenomenon on the contrary was not obvious. From this, we hypothesize that the development of resistance in BDQ usually occurs alongside resistance in CLO (32). It is suggested that the development and standardization of BDQ and CLO susceptibility test methods in this study can help to detect the emergence of cross drug resistance. The number of BDQ-resistant and CLO-resistant isolates in this study was limited; the numbers of isolates will be expanded for validation.

**Conclusion.** In this study, the MYCO test system of the Sensititre customized panel was used to comprehensively analyze the 8 repurposed anti-NTM antibacterial agents for NTM in Shanghai. Only BDQ and CLO exhibited effective activity against the employed SGM and RGM, and some *M. kansasii* strains were sensitive to DLM. This can be applied to the treatment of NTM diseases. According to the tentative ECOFF data in our assay, 0.5 µg/mL and 1 µg/mL could be tentatively proposed as breakpoints for NTM susceptibility testing for BDQ and CLO, respectively. Cross-resistance of BDQ and CLO can be detected by the Sensititre customized panel simultaneously.

## MATERIALS AND METHODS

**Ethical approval of the study protocol.** The study protocol was approved by the Ethics Committees of Shanghai Pulmonary Hospital, affiliated with Tongji University (K19-008), Shanghai, China. It was carried out in line with the ethical standards laid down in the 1964 Declaration of Helsinki and its later amendments. Patients provided written informed consent to have their data included in this study.

**Reference strains.** Reference strains of *S. aureus* (ATCC 29213), *E. coli* (ATCC 25922), *M. avium* (ATCC 700898) and H37RV (ATCC 27294) were used to perform MYCO test system.

**Isolated nontuberculous mycobacteria.** A total of 241 NTM clinical isolates, collected between November 2020 and October 2021 at Shanghai Pulmonary Hospital, affiliated with Tongji University, Shanghai, China, were analyzed in this study as shown in Fig. 1. These included 60 RGM, which were 46 isolates of *Mycobacterium abscessus* and 14 of other RGM, and 181 SGM, which were 122 isolates of *Mycobacterium intracellulare*, 24 of *Mycobacterium avium*, 21 of *Mycobacterium kansasii*, and 14 of other SGM. All NTM clinical strains were isolated from patients suspected of having tuberculosis by the proportion method using the Bactec MGIT 960 system. The strains were preliminarily classified as NTM using a *p*-nitrobenzoic acid-containing medium and were then identified at the species level by sequencing (Zeesan Biotech Co., Ltd.). All isolates were stored in 7H9 broth (Becton, Dickinson, Franklin Lakes, NJ) with 10% OADC (0.05% oleic acid, 5% bovine serum albumin [BSA], 2% dextrose, 0.004% catalase)

containing 15% glycerol in a freezer at −80°C until subcultured. These clinical isolates were grown at 37°C on Lowenstein-Jensen (LJ) medium (Baso) until grown to the mid-log phase (optical density at 590 nm [OD$_{590}$] of ∼0.4 [∼2.5 × 10$^8$ CFU/mL]), before being subjected to antimicrobial susceptibility testing (AST). Using an ultrasonic grinder (TB Healthcare, China), 0.5 McFarland bacterial suspensions were prepared from colonies grown on LJ medium and 50 $\mu$L bacterial suspension was added into 10 mL Middlebrook Mueller-Hinton broth (Becton Dickson) to a final concentration of ∼10$^5$ CFU/mL and used for the AST. The SGM test isolates were diluted and cultured in Middlebrook Mueller-Hinton broth supplemented with 10% OADC. The RGM test isolates followed the same procedure without addition of OADC.

**Antimicrobial agents.** The MYCO test system included the Sensititre SLOMYCO and RAPMYCOI panels (Trek Diagnostics/Thermo Fisher, Bremen, Germany) for testing the drug sensitivity of NTM strains (28, 33) and was prepared according to the manufacturer's instructions. For the 8 potential drugs, we used the same approach to design customized panels. This standard off-the-shelf AST panel format includes three different subpanels:

1. SLOMYCO panels with 13 antimicrobials for SGM: amikacin (AMK), ciprofloxacin (CIP), clarithromycin (CLA), doxycycline (DOX), ethambutol (EMB), ethyl ethionamide (ETH), isoniazid (INH), linezolid (LZD), moxifloxacin (MXF), rifampin (RIF), rifabutin (RFB), streptomycin (STR), and trimethoprim-sulfamethoxazole (SXT).
2. RAPMYCO panels with 15 antimicrobials for RGM: amikacin (AMK), ciprofloxacin (CIP), clarithromycin (CLA), doxycycline (DOX), linezolid (LZD), moxifloxacin (MXF), trimethoprim-sulfamethoxazole (SXT), amoxicillin-clavulanic acid (AMC), cefepime (FEP), cefoxitin (FOX), ceftriaxone (CRO), imipenem (IMP), minocycline (MIN), tigecycline (TGC), and tobramycin (TOB).
3. Customized panels with 8 antimicrobials for NTM: vancomycin (VAN), bedaquiline (BDQ), delamanid (DLM), faropenem (FAR), meropenem (MPM), clofazimine (CLO), cefoperazone-avibactam (CFP-AVI), and cefoxitin (FOX).

**Antimicrobial susceptibility testing.** For AST, 100 $\mu$L of an ∼10$^5$-CFU/mL NTM suspension was distributed into each well using an automatic dispenser. Incubated at 37°C, SGM and RGM were incubated for 7 and 3 days, respectively, to observe the growth of positive-control wells. Visible apparent bacterial precipitation at the bottom of the wells was considered positive. The MIC was determined as the lowest concentration of the drug that inhibited the visible growth of the isolates. SGM are reincubated for up to a total of 10 days should there be insufficient growth at 7 days, and RGM are reincubated for up to a total of 5 days should there be insufficient growth at 3 days.

Carbapenems (IMP, FAR, and MEM) are unstable over time, which if not properly controlled could lead to erroneous resistant results. Some SGM were cultured in carbapenem for 1 week for observation and recording, and the rest continued to culture. The MICs of all of the agents in the different panels were observed and recorded using the automated microbial susceptibility analysis system of Sensititre Vizion equipment. All of the drug results were read at 99% inhibition with the exception of SXT, which was read at 80% inhibition.

**Epidemiological cutoff value determination.** For species with sufficient isolates and excellent inhibitory activity as demonstrated by BDQ/CLO, ECOFFs were identified using ECOFFinder (EUCAST) (34). For the unimodal MIC distribution profile, the ECOFF was defined as the concentration that could inhibit 95% of the bacterial population.

**Statistical analysis.** Excel (MS Office 2019) software was used for data statistics, and GraphPad Prism 8 (San Diego, CA) was used for graphical display.

## SUPPLEMENTAL MATERIAL

Supplemental material is available online only.
**SUPPLEMENTAL FILE 1**, PDF file, 0.05 MB.

## ACKNOWLEDGMENTS

This project was supported by grants from the Shanghai Clinical Research Center for Infectious Diseases (TUBERCULOSIS) (19MC1910800 to W.S.).

We declare no conflict of interest.

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
