## [Reviewer comments · Microbiology Spectrum]

Microbiology Spectrum

Antimicrobial susceptibility testing using MYCO test-system and MIC distribution of 8 drugs against clinical isolates from Shanghai of Nontuberculous Mycobacteria

Ruo-yan Ying, Jinghui Yang, xiao-cui wu, Fangyou Yu, and Wei Sha

Corresponding Author(s): Wei Sha, Tongji University Affiliated Shanghai Pulmonary Hospital

Review Timeline:

Submission Date:	July 4, 2022
Editorial Decision:	October 16, 2022
Revision Received:	January 9, 2023
Accepted:	January 15, 2023

Editor: Xinchun Chen

Reviewer(s): Disclosure of reviewer identity is with reference to reviewer comments included in decision letter(s). The following individuals involved in review of your submission have agreed to reveal their identity: Bruno Lopes Abbadi (Reviewer #1)

Transaction Report:

DOI: <https://doi.org/10.1128/spectrum.02549-22>

October 16, 2022

Dr. Wei Sha
Shanghai Pulmonary Hospital, Tongji University School of Medicine
Clinic and Research Centre of Tuberculosis, Shanghai Key lab of Tuberculosis,
Shanghai
China

Re: Spectrum02549-22 (Antimicrobial susceptibility testing using MYCO test-system and MIC distribution of 8 drugs against clinical isolates from Shanghai of Nontuberculous Mycobacteria)

Dear Dr. Wei Sha:

Link Not Available

Sincerely,

Xinchun Chen

Journals Department
Reviewer comments:

Reviewer #1 (Comments for the Author):

Overview:

The present manuscript aims to evaluate the susceptibility of nontuberculous mycobacteria (fast- and slow-growing) to different drug panels. The authors came to the conclusion that fast- or slow-growing NTMs show varied susceptibility to the antibiotics tested, and that bedaquiline and clofazimine showed broad activity among different NTM species.

Major comments:

- 1) Line 71: Authors should include epidemiological data about NTM infectious to support their claim that those infections are increasing worldwide. The literature review in the introductory part is too short and general.
- 2) Line 72: What is the natural drug resistance for NTM? Authors should list the antibiotics to which NTMs are resistant.
- 3) Lines 72-75: What is the complex approach used to treat NTM? What are the most common treatments for these infections? What is the average treatment duration for these infections? What is the average percentile of efficacy? Authors should feed the introduction with more information, otherwise, it is too vague.
- 4) Lines 77-81: These sentences are unclear. The authors start by saying that "Based on the commonly used anti-NTM antibacterial drugs, 8 drugs were repurposed, namely... (list of drugs)...". Then, the second sentence says, "were selected in this study to evaluate drug sensitivity in vitro". Apparently, these two sentences are not linked. Also, why is bedaquiline and clofazimine considered to be repurposed drugs, if they are used to treat multidrug-resistant TB?
- 5) Line 82: I believe data is still limited. However, a quick search about this topic brings some insightful articles that could be included in the introduction or discussion sections. Below are examples of articles describing the use of the above-mentioned antibiotics against NTMs:

In vitro activity of bedaquiline and delamanid against nontuberculous mycobacteria, including macrolide-resistant clinical isolates (2019). Bedaquiline as a potential agent in the treatment of Mycobacterium abscessus infections (2017). Successful bedaquiline-containing antimycobacterial treatment in post-traumatic skin and soft-tissue infection by Mycobacterium fortuitum complex: a case report (2020). Emergence of nontuberculous mycobacteria infections during bedaquiline-containing regimes in multidrug-resistant tuberculosis patients. A bedaquiline/clofazimine combination regimen might add activity to the treatment of clinically relevant non-tuberculous mycobacteria (2019).

- 6) Line 106: Mycobacterium abscessus.
- 7) Lines 105-108: I would suggest creating a figure or graph to display all species/clinical isolates included in this study. Overall, the manuscript does not have many figures that illustrate the methodologies and or results raised, and the inclusion of this figure could make it easier to appreciate your work.
- 8) Lines 117-118: Poor writing quality sentence. I suggest rewriting it.
- 9) Lines 139-143: Overall, the writing quality of this paragraph is very low. I strongly suggest rewriting it to make the reading clearer and smoother for the reader.
- 10) Line 145: Authors wrote "it can be recorded and observed in advance". This sentence is not properly connected to the previous one. I suggest rewriting it.
- 11) Line 161: In Table 1 REB is RFB (rifampicin). I couldn't find what this abbreviation stands for in the text.
- 12) Lines 262-264: Overall, the writing quality of this paragraph is very low. Also, isn't there a specific code for each strain? There are 6 M. abscessus species, which I assume are different strains. Authors should find a better way to describe these strains.
- 13) Lines 270-271: Low-quality writing.
- 14) Line 275: Species are sensitive to a given antibiotic, not the antibiotics themselves.
- 15) Lines 288-289: Any data to support this sentence? What is the concept of extensive drug resistance here?
- 16) Lines 293-294: Low-quality writing.

Minor comments:

- 1) Authors should include a space between the MIC value and the unit (16 µg/ml). In addition, the unit ug/ml is unacceptable.
- 2) The name of all species should be in italics, even when they appear in tables and figures.

Reviewer #2 (Comments for the Author):

The study by Ying et al performed drug-susceptibility tests for 241 NTM isolates against antibiotics using a commercial system

(Sensititre SLOMYCO or RAPMYCOI) and an in-house assay. The commercial system included 13 or 15 antibiotics and the self-defined panel included 8 antibiotics. The NTM isolates were composed of different NTM species including *M. kansasii*, *M. avium*, *M. intracellulare* and *M. abscessus*. The authors reported that slow-growing NTMs had "good sensitivity" to amikacin, clarithromycin, and rifampin, while the rapidly growing NTM species only had sensitivity to amikacin and tigecycline. Bedaquiline and clofazimine had favorable activities against different NTM species. Similar studies have been done in other countries before (Carvalho, *Int J Mycobacteriol*, 2021; Litvinov, *PLoS One*, 2018), but such a dataset is scarce from China and would thus hold relevant significance for clinical reference.

Major comments

1. The overall impression is that the authors presented this work in a "raw data" format. There are too many tables but I didn't see the necessity of dividing them separately. Please try to combine those tables into one or two. Besides, please also consider reducing the raw text that just simply described the table contents - you don't need to repeat all the information that has already been presented in the table. Keep the text clear and concise, and highlight the major findings.
2. The methods section lacks a lot of details.
 - a. How were those clinical isolates processed before the AST experiments? Besides, for the AST experiment, the authors wrote "100 µl of NTM suspension was added to each panel". Did the authors measure the number of bacterial cells in the inoculation and normalize different isolates to similar input? This is important because different inoculations might affect the interpretation of AST results.
 - b. Eight drugs apart from those of the MYCO test system were selected to perform the antimicrobial susceptibility testing, namely, vancomycin (VA), bedaquiline (BDQ), delamanid (DLM), faropenem (FAR), meropenem (MPM), clofazimine (CFZ), avibactam (CAZ), and cefoxitin (FOX). Please provide the rationality of choosing the eight drugs.
 - c. How were the MIC values determined? The lowest concentration of the drug that inhibited the visible growth of the isolates tested? How did you measure visible growth? Please specify.
 - d. Please provide the details for epidemiological cut-off value (ECOFFs) determination of BDQ and CFZ, and please discuss why the ECOFFs values of BDQ differed from those adopted by Yu et al (*Antimicrob Agents Chemother*, 2019) (Line 295).
3. The authors concluded that "The increased MIC value of BDQ corresponds to the increased MIC value of CFZ", but the MIC values of *M. abscessus* against BDQ and CFZ as shown in Table 9 did not support this conclusion. Among the six *M. abscessus* isolates with MIC values of BDQ greater than the minimum MIC range, three (50%) with the MIC of 0.12 µg/ml, which is the minimum MIC value of *M. abscessus* isolates to CFZ.

Minor comments

1. This manuscript needs considerable language editing. For example, phrasing like "good sensitivity" is confusing.
1. Line 264, the authors stated nine of the eleven isolates with MIC values of CFZ increase to 1- to 3-fold, but the results shown in Figure 2 and Table 9 indicate six.
2. "REB" in the "MIC distribution of SGM species in SLOMYCO panel" section of results (Lines 159-187) should be "RFB".
3. Consider using figures to present Table 3 and Table 7.
4. All Latin names of species should be italic.
5. Reference 26 is incomplete, lacking volume, issue and pages.

Reviewer #3 (Comments for the Author):

1. some references are incomplete, such as
ref 17. Gail L. Woods M. 2018. M24 Susceptibility Testing of Mycobacteria, Nocardia spp., and Other Aerobic Actinomycetes, 3rd Edition, Third ed. the press needs to be provided.
ref 16. 16. Franklin R, Cockerill I. 2011. Performance Standards for Antimicrobial Susceptibility Testing. Twenty-first informational supplement M100-MS21.
2. Delamanid showed high MIC values for all NTM except *Mycobacterium kansasii*, this is mentioned in Kim, D. H., et al. (2019). "In Vitro Activity of Bedaquiline and Delamanid against Nontuberculous Mycobacteria, Including Macrolide-Resistant Clinical Isolates." *Antimicrob Agents Chemother* 63(8). The MS has no MIC value for other SGM, please supplement.
in the same reference, the author reported "bedaquiline showed low MIC values for the major pathogenic NTM, including *Mycobacterium avium* complex, *Mycobacterium abscessus* subsp. *abscessus*, *M. abscessus* subsp. *massiliense*, and *M. kansasii* Bedaquiline also had low MIC values with macrolide-resistant NTM strains", there is discrepancy in the MS for the bedaquiline, the authors can give some explanation.
3. delamanid (Dlm) was not active against most of the tested reference strains and clinical isolates of RGM species, as documented in Yu, X., et al. (2019). "In Vitro Activities of Bedaquiline and Delamanid against Nontuberculous Mycobacteria Isolated in Beijing, China." *Antimicrob Agents Chemother* 63(8). how to reconcile the results with this MS?
4. the number of strains used for test varies significantly, can the author explain the design logic for the imbalance NTM strain numbers?
5. there is no statistical design for the results.

The MS needs revision.

Staff Comments:

Preparing Revision Guidelines

Please return the manuscript within 60 days; if you cannot complete the modification within this time period, please contact me. If you do not wish to modify the manuscript and prefer to submit it to another journal, please notify me of your decision immediately so that the manuscript may be formally withdrawn from consideration by Microbiology Spectrum.

Reviewer comments:

Reviewer #1 (Comments for the Author):

Overview:

The present manuscript aims to evaluate the susceptibility of nontuberculous mycobacteria (fast- and slow-growing) to different drug panels. The authors came to the conclusion that fast- or slow-growing NTMs show varied susceptibility to the antibiotics tested, and that bedaquiline and clofazimine showed broad activity among different NTM species.

Major comments:

1) Line 71: Authors should include epidemiological data about NTM infectious to support their claim that those infections are increasing worldwide. The literature review in the introductory part is too short and general.

As per the reviewer's suggestion, we have made the correction accordingly.

2) Line 72: What is the natural drug resistance for NTM? Authors should list the antibiotics to which NTMs are resistant.

Most NTM are intrinsically resistant or only partially sensitive to first-line anti-tuberculous drugs.

3) Lines 72-75: What is the complex approach used to treat NTM? What are the most common treatments for these infections? What is the average treatment duration for these infections? What is the average percentile of efficacy? Authors should feed the introduction with more information, otherwise, it is too vague.

Most NTM are intrinsically resistant or only partially sensitive to first-line anti-tuberculous drugs. Unlike tuberculosis (TB), treatment for NTM disease takes very long period of time, with a regimen consisting of a macrolides as the core drug and 2-4 other antibiotics until 12 months after sputum conversion.

4) Lines 77-81: These sentences are unclear. The authors start by saying that "Based on the commonly used anti-NTM antibacterial drugs, 8 drugs were repurposed, namely... (list of drugs)...". Then, the second sentence says, "were selected in this study to evaluate drug sensitivity in vitro". Apparently, these two sentences are not linked. Also, why is bedaquiline and clofazimine considered to be repurposed drugs, if they are used to treat multidrug-resistant TB?

We thank the reviewer for the comments. As per the reviewer's suggestion, we have improved the description of the sentences. Under the situation that research and development of potent novel antibiotics specific to NTM is sluggish, evaluating the effects of existing antibiotics and new anti-tuberculosis drugs (here we named them repurposed drugs) on NTM is a feasible way to develop new therapeutic strategies for NTM-PD. Meanwhile, new anti-TB drugs such as bedaquiline, delamanid and clofazimine have been shown potent bacteriostatic activity against NTM both in vivo and in vitro. Below are examples of articles describing the use of the above-mentioned drugs against NTMs: In Vitro Bedaquiline and Clofazimine Susceptibility Testing in Mycobacterium abscessus(2022). In Vitro Activities of Bedaquiline and Delamanid against Nontuberculous Mycobacteria Isolated in Beijing, China(2019). Efficacy of Bedaquiline, Alone or in Combination with Imipenem, against Mycobacterium abscessus in C3HeB/FeJ Mice(2020).

5) Line 82: I believe data is still limited. However, a quick search about this topic

brings some insightful articles that could be included in the introduction or discussion sections. Below are examples of articles describing the use of the above-mentioned antibiotics against NTMs:

In vitro activity of bedaquiline and delamanid against nontuberculous mycobacteria, including macrolide-resistant clinical isolates (2019). Bedaquiline as a potential agent in the treatment of Mycobacterium abscessus infections (2017). Successful bedaquiline-containing antimycobacterial treatment in post-traumatic skin and soft-tissue infection by Mycobacterium fortuitum complex: a case report (2020).

Emergence of nontuberculous mycobacteria infections during bedaquiline-containing regimens in multidrug-resistant tuberculosis patients. A bedaquiline/clofazimine combination regimen might add activity to the treatment of clinically relevant non-tuberculous mycobacteria (2019).

We thank the reviewer for the comments. We have included these in the discussion section and cited these papers.

6) Line 106: Mycobacterium abscessus.

As per the reviewer's suggestion, we have made the correction accordingly.

7) Lines 105-108: I would suggest creating a figure or graph to display all species/clinical isolates included in this study. Overall, the manuscript does not have many figures that illustrate the methodologies and or results raised, and the inclusion of this figure could make it easier to appreciate your work.

As per the reviewer's suggestion, we have showed that in Fig 1.

8) Lines 117-118: Poor writing quality sentence. I suggest rewriting it.

As per the reviewer's suggestion, we have made the correction accordingly.

9) Lines 139-143: Overall, the writing quality of this paragraph is very low. I strongly suggest rewriting it to make the reading clearer and smoother for the reader.

As per the reviewer's suggestion, we have made the correction accordingly.

10) Line 145: Authors wrote "it can be recorded and observed in advance". This sentence is not properly connected to the previous one. I suggest rewriting it.

As per the reviewer's suggestion, we have made the correction accordingly.

11) Line 161: In Table 1 REB is RFB (rifampicin). I couldn't find what this abbreviation stands for in the text.

As per the reviewer's suggestion, we have made the correction accordingly.

12) Lines 262-264: Overall, the writing quality of this paragraph is very low. Also, isn't

there a specific code for each strain? There are 6 *M. abscessus* species, which I assume are different strains. Authors should find a better way to describe these strains.

As per the reviewer's suggestion, we have rewritten this paragraph. Because our mycobacterium identification kit does not support subdivision into subspecies, so our study focused on the four predominant prevalent species.

13) Lines 270-271: Low-quality writing.

As per the reviewer's suggestion, we have made the correction accordingly.

14) Line 275: Species are sensitive to a given antibiotic, not the antibiotics themselves.

We thank the reviewer for the comments. As per the reviewer's suggestion, we have improved the description of the sentences.

15) Lines 288-289: Any data to support this sentence? What is the concept of extensive drug resistance here?

As per the reviewer's suggestion, we have improved the description of the sentences. Treatment of RGM infection is very difficult because this bacterium is resistant to a wider variety of drugs than SGM. According to the reference 30(Guz L, Puk K. 2022), they are reach show that the drug sensitivity of NTM varies from species to species. KAN, AMK, CLR and SMX were the most active against RGM isolates, and these same four plus DOX and CIP were the best drugs against SGM isolates.

16) Lines 293-294: Low-quality writing.

As per the reviewer's suggestion, we have made the correction accordingly.

Minor comments:

1) Authors should include a space between the MIC value and the unit (16 µg/ml). In addition, the unit ug/ml is unacceptable.

As per the reviewer's suggestion, we have made the correction accordingly.

2) The name of all species should be in italics, even when they appear in tables and figures.

As per the reviewer's suggestion, we have made the correction accordingly.

Reviewer #2 (Comments for the Author):

The study by Ying et al performed drug-susceptibility tests for 241 NTM isolates against antibiotics using a commercial system (Sensititre SLOMYCO or RAPMYCOI) and an in-house assay. The commercial system included 13 or 15 antibiotics and the self-defined panel included 8 antibiotics. The NTM isolates were composed of different NTM species including *M. kansasii*, *M. avium*, *M. intracellulare* and *M. abscessus*. The authors reported that slow-growing NTMs had "good sensitivity" to amikacin, clarithromycin, and rifampin, while the rapidly growing NTM species only had sensitivity to amikacin and tigecycline. Bedaquiline and clofazimine had favorable

activities against different NTM species. Similar studies have been done in other countries before (Carvalho, Int J Mycobacteriol, 2021; Litvinov, PLoS One, 2018), but such a dataset is scarce from China and would thus hold relevant significance for clinical reference.

Major comments

1. The overall impression is that the authors presented this work in a "raw data" format. There are too many tables but I didn't see the necessity of dividing them separately. Please try to combine those tables into one or two. Besides, please also consider reducing the raw text that just simply described the table contents - you don't need to repeat all the information that has already been presented in the table. Keep the text clear and concise, and highlight the major findings.

We thank the reviewer for the comments. As per the reviewer's suggestion, we have improved the description of the results obtained.

2. The methods section lacks a lot of details.

a. How were those clinical isolates processed before the AST experiments? Besides, for the AST experiment, the authors wrote "100 µl of NTM suspension was added to each panel". Did the authors measure the number of bacterial cells in the inoculation and normalize different isolates to similar input? This is important because different inoculations might affect the interpretation of AST results.

a. All isolates were stored in 7H9 broth (Becton Dickinson, Franklin Lakes, NJ) containing 15% glycerol in a freezer at -80°C until subcultured. These clinical isolates were grown at 37°C on Lowenstein-Jensen (LJ) medium

(Baso) until growth to mid-log phase ($OD_{590} \approx 0.4$, $\sim 2.5 \times 10^8$ CFU/mL), before being subjected to antimicrobial susceptibility testing (AST). The last sentence in the section on Isolated nontuberculous mycobacteria, using an ultrasonic grinder (TB Healthcare, China), 0.5 McFarland bacterial suspension were prepared from colonies grown on L-J medium and add 50 μ l bacterial suspension into 10 ml Middlebrook Mueller–Hinton broth (Becton Dickson) to a final concentration of $\sim 10^5$ CFU/mL and used for the AST. The SGM test isolates were diluted and cultured in Middlebrook Mueller Hinton broth supplemented with 10% ADC [5% bovine serum albumin (BSA), 2% dextrose, 5% catalase]. The RGM test isolates did the same performance without adding ADC.

b. Eight drugs apart from those of the MYCO test system were selected to perform the antimicrobial susceptibility testing, namely, vancomycin (VA), bedaquiline (BDQ), delamanid (DLM), faropenem (FAR), meropenem (MPM), clofazimine (CFZ), avibactam (CAZ), and ceftiofloxacin (FOX). Please provide the rationality of choosing the eight drugs.

b. Under the situation that research and development of potent novel antibiotics specific to NTM is sluggish, evaluating the effects of existing antibiotics and new anti-tuberculosis drugs (here we named them *repurposed drugs*) on NTM is a feasible way to develop new therapeutic strategies for NTM-PD. NTM are gram-positive, aerobic bacteria, belonging same genus as *Mycobacterium tuberculosis* (MTB). β -lactam antibiotics are introduced to treat some NTM infection like carbapenem with or without β -lactamase. Recent years, new generation of β -lactam antibiotics and oral preparations have attracted attention since pre-clinical studies and case reports presented promising efficacy in treating infections due to NTM. Meanwhile, new anti-TB drugs such as bedaquiline, delamanid and clofazimine have been shown potent bacteriostatic activity against NTM both in vivo and in vitro. Below are examples of articles describing the use of the above-mentioned drugs against NTMs:

British Thoracic Society guidelines for the management of non-tuberculous mycobacterial pulmonary disease (NTM-PD)(2017). Diagnosis and treatment guidelines for nontuberculous mycobacterial diseases(2020). The synergistic effect of Imipenem-clarithromycin combination in the *Mycobacteroides abscessus*

complex(2020). Mycobacterium abscessus pulmonary disease: individual patient data meta-analysis(2019). Effect of Amoxicillin in combination with Imipenem-Relebactam against Mycobacterium abscessus(2020). β -Lactamase inhibition by avibactam in Mycobacterium abscessus(2015). Cutaneous Mycobacterium chelonae infection successfully treated with faropenem(2011). In Vitro Bedaquiline and Clofazimine Susceptibility Testing in Mycobacterium abscessus(2022). In Vitro Activities of Bedaquiline and Delamanid against Nontuberculous Mycobacteria Isolated in Beijing, China(2019). Efficacy of Bedaquiline, Alone or in Combination with Imipenem, against Mycobacterium abscessus in C3HeB/FeJ Mice(2020).

c. How were the MIC values determined? The lowest concentration of the drug that inhibited the visible growth of the isolates tested? How did you measure visible growth? Please specify.

c. Observe the growth of positive control wells , visible apparent bacterial precipitation at the bottom of the wells were considered positive. The minimum inhibitory concentration (MIC) was determined the lowest concentration of the drug that inhibited the visible growth of the isolate tests.

d. Please provide the details for epidemiological cut-off value (ECOFFs) determination of BDQ and CFZ, and please discuss why the ECOFFs values of BDQ differed from those adopted by Yu et al (Antimicrob Agents Chemother, 2019) (Line 295).

d. The ECOFF was determined according to the distribution profile of the MIC values using ECOFFinder (<https://clsi.org/meetings/susceptibility-testing-subcommittees/ecoffinder/>). In our study, the MIC range was set narrowly, with a large amount of collected data and a concentrated distribution of results. In Yu et al study, the MIC range was set broadly, with a small amount of collected data and a scattered distribution of results.

3. The authors concluded that "The increased MIC value of BDQ corresponds to the increased MIC value of CFZ", but the MIC values of *M. abscessus* against BDQ and CFZ as shown in Table 9 did not support this conclusion. Among the six *M. abscessus* isolates with MIC values of BDQ greater than the minimum MIC range, three (50%) with the MIC of 0.12 µg/ml, which is the minimum MIC value of *M. abscessus* isolates to CFZ.

We detected an increase in MIC values for BDQ in 11 strains, followed by an increase in MIC values of CFZ (9/11), as shown in Table 7. Meanwhile, the same phenomenon on the contrary was not obvious. From this, we hypothesize that the development of resistance in BDQ usually commonly occur alongside resistance in CFZ

The minimum MIC value of *M. abscessus* isolates to CFZ was ≤ 0.06 .

Minor comments

1. This manuscript needs considerable language editing. For example, phrasing like "good sensitivity" is confusing.

We thank the reviewer for the comments. As per the reviewer's suggestion, we have improved the description of the sentences.

1. Line 264, the authors stated nine of the eleven isolates with MIC values of CFZ increase to 1- to 3-fold, but the results shown in Figure 2 and Table 9 indicate six.

We didn't find this sentence at Line 264. We think it should be Line 277. Among the three most prevalent SGM (*M. kansasii*, *M. avium* and *M. intracellulare*), compared with the levels for *M. kansasii*, the MICs of more than half of the strains increased 1- to 2-fold for *M. avium* and *M. intracellulare*, as shown in Figure 5.

The results in Table 7 showed among the 241 patients, we detected an increase in MIC values for BDQ in 11 strains, followed by an increase in MIC values of CFZ (9/11).

2. "REB" in the "MIC distribution of SGM species in SLOMYCO panel" section of results (Lines 159-187) should be "RFB".

As per the reviewer's suggestion, we have made the correction accordingly.

3. Consider using figures to present Table 3 and Table 7.

As per the reviewer's suggestion, we have made the correction accordingly.

4. All Latin names of species should be italic.

As per the reviewer's suggestion, we have made the correction accordingly.

5. Reference 26 is incomplete, lacking volume, issue and pages.

As per the reviewer's suggestion, we have made the correction accordingly.

Reviewer #3 (Comments for the Author):

1. some references are incomplete, such as

ref 17. Gail L. Woods M. 2018. M24 Susceptibility Testing of Mycobacteria, Nocardia spp., and Other Aerobic Actinomycetes, 3rd Edition, Third ed. the press needs to be provided.

ref 16. 16. Franklin R, Cockerill I. 2011. Performance Standards for Antimicrobial Susceptibility Testing. Twenty-first informational supplement M100-MS21.

As per the reviewer’s suggestion, we have made the correction accordingly.

2. Delamanid showed high MIC values for all NTM except *Mycobacterium kansasii*, this is mentioned in Kim, D. H., et al. (2019). "In Vitro Activity of Bedaquiline and Delamanid against Nontuberculous Mycobacteria, Including Macrolide-Resistant Clinical Isolates." *Antimicrob Agents Chemother* 63(8). The MS has no MIC value for other SGM, please supplement.

in the same reference, the author reported "bedaquiline showed low MIC values for the major pathogenic NTM, including *Mycobacterium avium* complex, *Mycobacterium abscessus* subsp. *abscessus*, *M. abscessus* subsp. *massiliense*, and *M. kansasii*. Bedaquiline also had low MIC values with macrolide-resistant NTM strains", there is discrepancy in the MS for the bedaquiline, the authors can give some explanation.

We have attached the MIC values of other SGM and RGM for Bdq and Cfz in the supplementary materials. However, Dlm did not show a very low MIC value in our study, there were 21 strains of *M. kansasii* in our study, and one third (7/21) exceeded the set maximum MIC value, as show as in Table 3. In the study of Kim, D. H., et al, there were 100 strains, and only 2 strains exceeded the maximum MIC value. As can be seen in the table below, among other SGM, only two strains, one strain of *M. kansasii* and *M. avium*, and one strain of *M. szulgai*, were less than the maximum MIC value.

Category	species	No. of	MIC(s) (g/ml) by antimicrobial agent (no. of isolates)		
			Bdq	Cfz	Dlm

		isolates			
SGM	M.intracellulare and				
	M.avium	1	<=0.12	0.12	>0.5
	M.intracellulare and other				
	mycobacteria	1	<=0.12	<=0.06	>0.5
	M.kansasii and				
	M.intracellulare	1	<=0.12	0.12	>0.5
	M.kansasii and M.avium	1	<=0.12	<=0.06	0.25
	M.malmoense and and				
	M.avium	1	<=0.12	<=0.06	>0.5
	M.malmoense and other				
	mycobacteria	1	<=0.12	0.12	>0.5
	M.gordonae	1	<=0.12	0.5	>0.5
	M.flavescens	1	<=0.12	<=0.06	>0.5
	M.szulgai	2	<=0.12 (2)	<=0.06 (2)	0.25, >0.5
other mycobacteria	4	<=0.12 (4)	<=0.06 (3), 0.12	>0.5 (4)	
FGM	M.abscessus and			0.12 (4), 0.25	
	M.intracellulare	8	<=0.12 (7), 0.25	(4)	>0.5 (8)
	M.abscessus and				
	M.avium	2	<=0.12 (2)	0.12, 0.25	>0.5 (2)
	M.intracellulare and other				
mycobacteria	1	>4	0.5	>0.5	

M,fortuitum	2	<=0.12 (2)	0.12, 0.25	>0.5 (2)
other mycobacteria	1	<=0.12	0.25	>0.5

In a study by Kim, D. H., et al., a subtotal type for *M. abscessus* and a MAC analysis were performed and come to a conclusion that including *Mycobacterium avium complex*, *Mycobacterium abscessus subsp. abscessus*, *M. abscessus subsp. massiliense*, and *M. kansasii* Bedaquiline also had low MIC values with macrolide-resistant NTM strains. In our research results section, because our mycobacterium identification kit does not support subdivision into subspecies, so our study focused on the four predominant prevalent species. the sensitivity of *M. intracellulare*, *M. avium* and *M. kansasii* to CLA (macrolide antibiotics) was 95.9% (117/122), 100% (24/24) and 95.2% (20/21), respectively, *M. abscessus* had a sensitivity of 39.1% (18/46) to CLA, however, BDQ showed very low MIC in four prevalent NTM, and only one strain *M. intracellulare* and one strain *M. kansasii* had MIC values higher than the maximum.

3.delamanid (Dlm) was not active against most of the tested reference strains and clinical isolates of RGM species,as documented in Yu, X., et al. (2019). "In Vitro Activities of Bedaquiline and Delamanid against Nontuberculous Mycobacteria Isolated in Beijing, China." Antimicrob Agents Chemother 63(8). how to reconcile the results with this MS?

Dlm may be usable for some SGM species infection treatment but not very promising for RGM infection treatment, as documented in Yu, X., et al. Our study confirmed part of the content, indeed Dlm did not have a strong bacteriostatic effect on RGM, among the four common SGM, only partial *M. kansasii* (14/21) were inhibited.

4. the number of strains used for test varies significantly, can the author explain the design logic for the imbalance NTM strain numbers?

In this regression study, 241 clinical isolates of NTM patients admitted to Shanghai Pulmonary Hospital from 2020 to 2021 were collected. The distribution of NTM species usually has regional characteristics. *Mycobacterium intracellulare*

was most isolated in clinical specimens, and *Mycobacterium abscessus* also accounted for a higher proportion.

5. there is no statistical design for the results.

Statistical design has been added in compliance with the reviewers' comments.

Excel (MS Office 2019) software was used for data statistics and GraphPad Prism 8 (San Diego, CA) was used for graphical display.

The MS needs revision.

January 15, 2023

Dr. Wei Sha
Tongji University Affiliated Shanghai Pulmonary Hospital
Clinic and Research Center of Tuberculosis, Shanghai Key Laboratory of Tuberculosis, Shanghai Pulmonary Hospital, Tongji
University School of Medicine
Shanghai
China

Re: Spectrum02549-22R1 (Antimicrobial susceptibility testing using MYCO test-system and MIC distribution of 8 drugs against clinical isolates from Shanghai of Nontuberculous Mycobacteria)

Dear Dr. Wei Sha:

Pls add more detail legend to each figure.

Your manuscript has been accepted, and I am forwarding it to the ASM Journals Department for publication. You will be notified when your proofs are ready to be viewed.

Sincerely,

Xinchun Chen
Editor, Microbiology Spectrum
